# Associations among weed communities, management practices, and environmental factors in U.S. snap bean (*Phaseolus vulgaris*) production

Pavle Pavlovic[1¤a] *, Jed B. Colquhoun[2], Nicholas E. Korres[3¤b], Christopher A. Landau[3], Rui Liu[4], Carolyn J. Lowry[5], Nicolas F. Martin[1], Ed Peachey[6], Barbara Scott[7], Lynn M. Sosnoskie[8], Mark J. VanGessel[7], Martin M. Williams II[3]

1 Department of Crop Sciences, University of Illinois Urbana-Champaign, Urbana, Illinois, United States of America, 2 Department of Plant and Agroecosystem Sciences, University of Wisconsin-Madison, Madison, Wisconsin, United States of America, 3 United States Department of Agriculture, Agricultural Research Service, Global Change and Photosynthesis Research Unit, Urbana, Illinois, United States of America, 4 Department of Crop and Soil Sciences, Washington State University, Pullman, Washington, United States of America, 5 Department of Plant Science, the Pennsylvania State University, University Park, Pennsylvania, United States of America, 6 Department of Horticulture, Oregon State University, Corvallis, Oregon, United States of America, 7 Department of Plant and Soil Sciences, University of Delaware, Georgetown, Delaware, United States of America, 8 School of Integrative Plant Science – Horticulture Section, Cornell AgriTech, Cornell University, Geneva, New York, United States of America

¤a Current address: Department of Soil and Crop Sciences, Texas A&M University, College Station, TX, USA
¤b Current address: Department of Agriculture, University of Ioannina, Ioannina, Greece
* pavlep@tamu.edu

## Abstract

Weed species that escape control (hereafter called residual weeds) coupled with changing weather patterns are emerging challenges for snap bean processors and growers. Field surveys were conducted to identify associations among crop/weed management practices and environmental factors on snap bean yield and residual weed density. From 2019–2023, a total of 358 snap bean production fields throughout the major U.S. production regions (Northwest, Midwest and Northeast) were surveyed for residual weeds. Field-level information on crop/weed management, soils, and weather also were obtained. To determine associations among management and environmental variables on crop yield and residual weed density, the machine learning algorithm random forest was utilized. The models had 24 and 22 predictor variables for crop yield and residual weed density, respectively, and both were trained on 80% of the data with the remainder used as a test set to determine model accuracy. Both models had pseudo-$R^2$ values of over 0.50 and accuracy over 80%. The models showed that crop yield was higher in the Northwest compared to the Midwest region, while higher average temperatures during early season growth and planting mid-season (June-July) predicted greater crop yield compared to other time periods. The use of row cultivation was associated with lower snap bean yield and weed density,

**Data availability statement:** All relevant data are within the manuscript and/or its Supporting Information files.

**Funding:** This research was funded by U.S. Department of Agriculture–Agricultural Research Service project 5012-12220-010-000D ("Resilience of Integrated Weed Management Systems to Climate Variability in Midwest Crop Production Systems"). Any opinions, findings, conclusions, or recommendations expressed in this publication are those of the authors and do not necessarily reflect the views of the U.S. Department of Agriculture. The mention of trade names or commercial products in this publication is solely for the purpose of providing specific information and does not imply recommendation or endorsement. The U.S. Department of Agriculture is an equal opportunity provider and employer.

**Competing interests:** The authors have declared that no competing interests exist.

suggesting row cultivation had less-than-ideal selectivity between the crop and weed. Moreover, multiple spring tillage operations prior to planting were linked with an increase in weed density, implying that excessive tillage may favor the emergence of residual weeds in snap bean. Over the coming decades, climate change-driven weather variability is likely to influence snap bean production, both directly through crop growth and indirectly through weeds that escape control practices that also are influenced by the weather.

## Introduction

Snap bean is a vegetable crop grown for its unripe fruits (pods), representing different cultivars of common bean (*Phaseolus vulgaris* L.). In the United States (US), commercial snap bean production is mainly for the processing market (~83% of overall production) and, to a lesser extent, the fresh market [1]. Processed snap bean is canned (two-thirds) and frozen (one-third) [2]. In the last five years, overall snap bean production tonnage has decreased by ~30% [2]. The reasons for declining production are two-fold: an increase in imported snap bean products and changing consumer preference towards fresh and frozen, rather than canned, products [1].

Snap bean growers are adjusting to a shifting market and increasing demand for frozen products. Snap bean is one of the more profitable crops in row crop rotation [3,4]; however, the crop is susceptible to drought, elevated temperatures during flowering (anthesis), and pests. Another significant threat to snap bean production is weed competition which can cause up to 80% yield losses [5,6]. In addition to yield reduction, many weeds can interfere with mechanical harvest and weed organs can contaminate harvested products and elude the sorting process [4,7]. Many snap bean processors have an extremely low tolerance for any kind of weed contamination and entire harvested loads can be rejected. To manage weeds in snap bean, growers rely, heavily, on herbicide applications [4]. However, the limited number of products registered for snap bean often leads to overreliance on a few herbicides or modes of action [4]. This overreliance can lead to resistance to one or more herbicide modes of action in certain weed species making their management in snap bean very challenging. Cultivation can also be used in snap bean; however, the successful integration and use of soil disturbance into a production system can be negatively impacted by adverse weather events (e.g., heavy rainfalls) and edaphic conditions (e.g., rocky soils). Labor is costly and becoming increasingly more difficult to source, preventing the widespread use of hand weeding [4].

Climate change is a major threat to crop production. Climate variability and the frequency, magnitude, and duration of weather extremes are on the rise [8,9]. Rapid transitions between precipitation extremes coupled with heat waves lasting longer and having a more severe effect on plant growth and development are expected to become more common in the future, major snap bean production regions included [10–12]. This will negatively affect all crop production, with specialty crops, like snap bean, being particularly threatened in the coming decades [13]. Snap bean is

sensitive to high temperatures during anthesis, resulting in yield losses due to flower abortion and pod abscission [14,15]. Extreme precipitation events, especially early season, can result in soil crusting and the proliferation of soil-borne pathogens, leading to seedling mortality [3,16]. Changing weather patterns may also result in differential growth responses between crops and weeds, making certain weeds more competitive, and reducing the efficacy of pre-emergence (PRE) and postemergence (POST) herbicides [17,18].

To improve the resiliency of US snap bean production, particularly with respect to weed control, it is crucial that we understand the drivers influencing weed density and diversity across the major cropping regions. The objective of this study was to identify the most important management practices and environmental factors related to the weed community and snap bean yield.

## Materials and methods

The overall approach was to survey snap bean production fields near the time of harvest to obtain data on the weed communities that escaped control (i.e., residual weeds). Between 2019 and 2023, collaborating vegetable processors provided lists of fields scheduled for harvest from which samples were drawn. There were no permits required for accessing fields in this research as all the fields surveyed were managed by vegetable processors, who grew them under contract from actual farm owners. Collaborating vegetable processors and individuals involved requested anonymity on this matter. Ultimately, 358 fields grown under contract for snap bean were surveyed in the three primary regions of processed snap bean production: specifically, the Northwest (NW), the Midwest (MW), and the Northeast (NE) U.S. [19]. Fields were surveyed in the states of Oregon and Washington in the NW (71 fields), Illinois, Iowa, Minnesota, and Wisconsin in the MW (205 fields), and Delaware, Maryland, New York, and Pennsylvania in the NE (82 fields) [20]. Surveys were conducted across a broad window of snap bean harvest from June to October.

Four types of data were obtained for each surveyed field: (1) weed community data, (2) crop and weed management data, (3) soil data, and (4) weather data. The methodology for surveying the weed community in each field was conducted by utilizing an approach previously described [21] with slight modifications that considered the weed distribution (patchiness) in the field. With respect to weed community data, researchers counted the numbers of weeds, per species or species group, in 30 quadrats per field [19]. Quadrats ranged from 0.5 m² (0.5m x 1m) to 1 m² (1m x 1m) in size. The exception was for fields less than 20 ha in size, where one quadrat of data was collected for each ha in snap bean production. These data were used to calculate average weed density for each field. In addition, % weed cover was also determined visually by species or species group [19]. Field sizes and the geolocation of surveyed fields were provided by the collaborating vegetable processors. Crop and weed management data consisted of variables characterizing agronomic practices and weed control programs in each surveyed field. Agronomic practices include the type and timing of tillage operations, snap bean variety planted, planting date, seeding rate and row width at the planting, the crop preceding snap bean, and utilization of irrigation. Weed control programs included the utilization of hand weeding (mechanical weed control using human labor), row cultivation (mechanical weed control using machinery), and chemical control (time of herbicide application, herbicide active ingredients, and modes of action used). Chemical control consists of different timings of herbicide application (preemergence [PRE] before crop emergence and postemergence [POST] after crop emergence), herbicide active ingredient which is the actual part of the herbicide product affecting weeds and herbicide mode of actions which represents how the active ingredient affects the plant on a physiological level once absorbed by it.

All agronomic data was provided at the discretion of the growers at the end of the growing season. For each field, 10 soil samples were collected to a depth of 20 cm using a 2.5 cm diameter soil probe, aggregated by field, and then analyzed for soil physical and chemical properties (A&L Great Lakes Laboratories, Inc., 3505 Conestoga Dr, Fort Wayne, IN 46808). These are usual properties analyzed by soil labs, such as pH, texture, organic matter or cation contents. For our analysis we used only some of these properties (as shown in Table 1), due to high correlation and confounding level between multiple properties. Weather data was obtained from the Daymet daily surface weather database on a 1-km grid

**Table 1. Predictor (management and environmental) variables used for random forest model building with their acronyms in the brackets.**

| Management Predictors | Environmental Predictors |
|---|---|
| Field Size (ha) [FS] | Organic Matter Content (%) [OM] |
| Seeding Rate (Seeds/ha) [SR] | pH Value [pH] |
| Planting Date (Julian Day) [PDN] | Sand Content (%) [SAND] |
| Row Width (cm) [RW] | Precipitation in Growth Interval I (first 20 days after planting) [mm] [PREC1] |
| Number of Spring Tillage Operations [STN] | Precipitation in Growth Interval II (21–40 days after planting) [mm] [PREC2] |
| Region | Precipitation in Growth Interval III (41 days after planting to harvest) [mm] [PREC3] |
| Snap Bean Variety [SBV] | Average Temperature in Growth Interval I (first 20 days after planting) [°C] [AVGT1] |
| Preceding Crop [PC] | Average Temperature in Growth Interval II (21–40 days after planting) [°C] [AVGT2] |
| Irrigation (Yes/No) [I] | Average Temperature in Growth Interval III (41 days after planting to harvest) [°C] [AVGT3] |
| Hand Weeding (Yes/No) [HW] | Weed Density (Plants/m2) [WD][a] |
| Row Cultivation (Yes/No) [RC] | Weed Cover (%) [WC][a] |
| Herbicide Application [HA] | |
| Herbicide Mode of Action Combinations [MoA] | |

[a] Weed Density and Weed Cover were used as predictor variables only for Crop Yield response

for North America by utilizing geospatial coordinates of each field [22]. Weather data included daily maximum and minimal air temperature and total daily precipitation from snap bean planting to harvest. Total precipitation and average daily temperatures were calculated for three intervals: first 20 days after planting, 21–40 days after planting, and 41 days to harvest (~60 days after planting).

The random forest regression was utilized to identify linkages among weed communities, management practices, and environmental factors [23]. Random forest is a collection of tree-structured models where each tree casts a unit vote for the most popular class of input. Because random forest is nonparametric, classical regression assumptions relating to data structure and distribution are not required. This modeling approach has been used previously in similar research on sweet corn [24] and it was evaluated that it was the most parsimonious approach. The fitted models used snap bean yield (Mt ha$^{-1}$) and average residual weed density (plants/m$^2$) as the response variables, predicted by 13 management variables and 11 or 9 environmental variables for crop yield and weed density, respectively (Table 1). Due to missing data, the number of fields analyzed for crop yield and weed density was 268 (only NW and MW regions) and 317 (all regions), respectively.

The package *ranger* in R statistical software version 4.4.0 [25] was used. Tuning parameters were set so that the model with the lowest root mean square error (RMSE) and highest goodness of fit (pseudo-$R^2$) values could be fitted. The number of individual regression trees was set to 1,000 as previously suggested [23]. The optimal number of independent variables randomly selected as candidates for each split in the trees was set to 13 and 12 for crop yield and weed density model, respectively. The minimum optimal number of observations in each terminal node was set to 5 for both models. For training both models, 80% of the fields were used, while the remaining 20% were used for checking the accuracy of the trained model. Prior to fitting the crop yield random forest model, the values of all numerical variables were scaled. For the weed density model, the values of all numerical variables were both scaled and transformed. Scaling was done utilizing Min-Max Scaling, while transformation was done utilizing Yeo-Johnson transformation same as in previously published

work [19]. Certain aspects of the algorithm are randomized; therefore, it is only possible to determine how important certain predictors were in the model, not necessarily what kind of relationship they have with the response variable. This predictor variable "importance" is defined as permutation importance, which considers the positive effect that each predictor had on the prediction performance [23]. The obtained regression models were used to predict crop yield and weed density based on their respective test sets. Predicted values were then correlated with their actual values from the test sets to obtain the accuracy of the models. Partial dependence plots of the most important predictor variables were used to visualize the rescaled and back-transformed response variables. These plots marginalize over the other predictors, thus giving a function that depends only on one or two chosen predictors from the model.

## Results and discussion

This study used a dataset built from on-farm surveys of snap bean production fields across the U.S. to identify the most important management practices and environmental factors linked to weed and crop outcomes. Random forest models had similar pseudo-$R^2$ values of 0.52 (±0.07) and 0.55 (±0.07) for crop yield (Fig 1A) and residual weed density (Fig 1B), respectively. The accuracy was 83.1% for the crop yield model and 80.6% for the residual weed density model. The model performance was within a range similar to previous research on sweet corn [25] and the effect of weather on weed control in soybean [17].

### Crop yield model

Predicting crop yield is a fundamental research question in plant biology resulting from complex interactions of crop genotype (G), the environment (E) and management practices (M), or the G x E x M paradigm [26]. Random forest has been

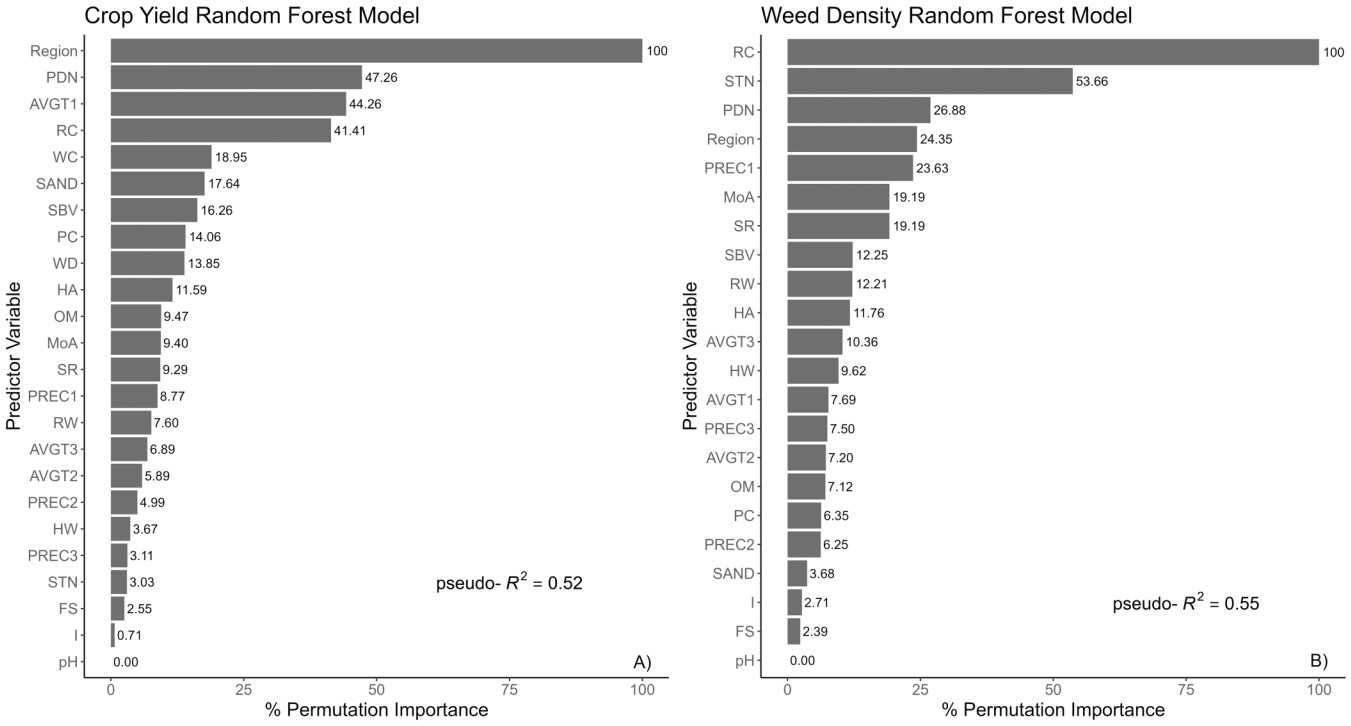

**Fig 1. Random Forest models of A) crop yield and B) weed density at harvest with % permutation importance of each of the predictor variables in their prediction accuracy.**

used to successfully model crop yield in agronomic production systems [27,28]. In our research, the fitted model displays this paradigm in the form of the most important predictor being the effect of the region where snap bean is grown (Fig 1A). Other variables that had importance in the model were: planting date as Julian day (PDN), the use of row cultivation (RC), and average temperature in growth interval I (AVGT1).

The NW region had a higher yield compared to the MW region (Fig 2A). The average yield for the MW states of Wisconsin, Minnesota, Michigan and Illinois was 10.6 Mt ha$^{-1}$, while the average yield for NW state of Oregon was 12.1 Mt ha$^{-1}$ according to the national data for 2023 [2]. Therefore, survey results align with the national data by demonstrating higher yield in the NW compared to the MW as demonstrated in previously published work [20]. The previously collected data [20] demonstrated that the management practices utilized in the crop production were very different between the two regions, as well as varieties grown in each region respectively. Also, the two regions had substantially different soil characteristics (Fig 3) in terms of their texture, as NW soils tended to be less sandy compared to MW soils. The climate differs between the two regions as MW is characterized by humid continental climate (Köppen climate types *Dfa*) with temperatures that vary greatly from summer to winter and appreciable precipitation [29], while the NW is characterized by warm-summer Mediterranean climate (Köppen climate type *Csb*), characterized by warm and dry summers, and mild to cool and wet winters [30]. The lack of precipitation in NW region allows for a better control of water supply on the heavier soils. This is important when applying PRE herbicides, as supply of water (either through precipitation or irrigation) allows for better incorporation of chemicals, increasing their effectiveness on weed populations. Arid climate conditions also make it more difficult for pathogens to proliferate, possibly allowing for narrower rows (and therefore higher seeding rates [20]) as the moisture in the crop canopy is controlled by irrigation rather than indiscriminate precipitation events. All these factors together are possible contributors to higher average crop yield in the NW compared to MW.

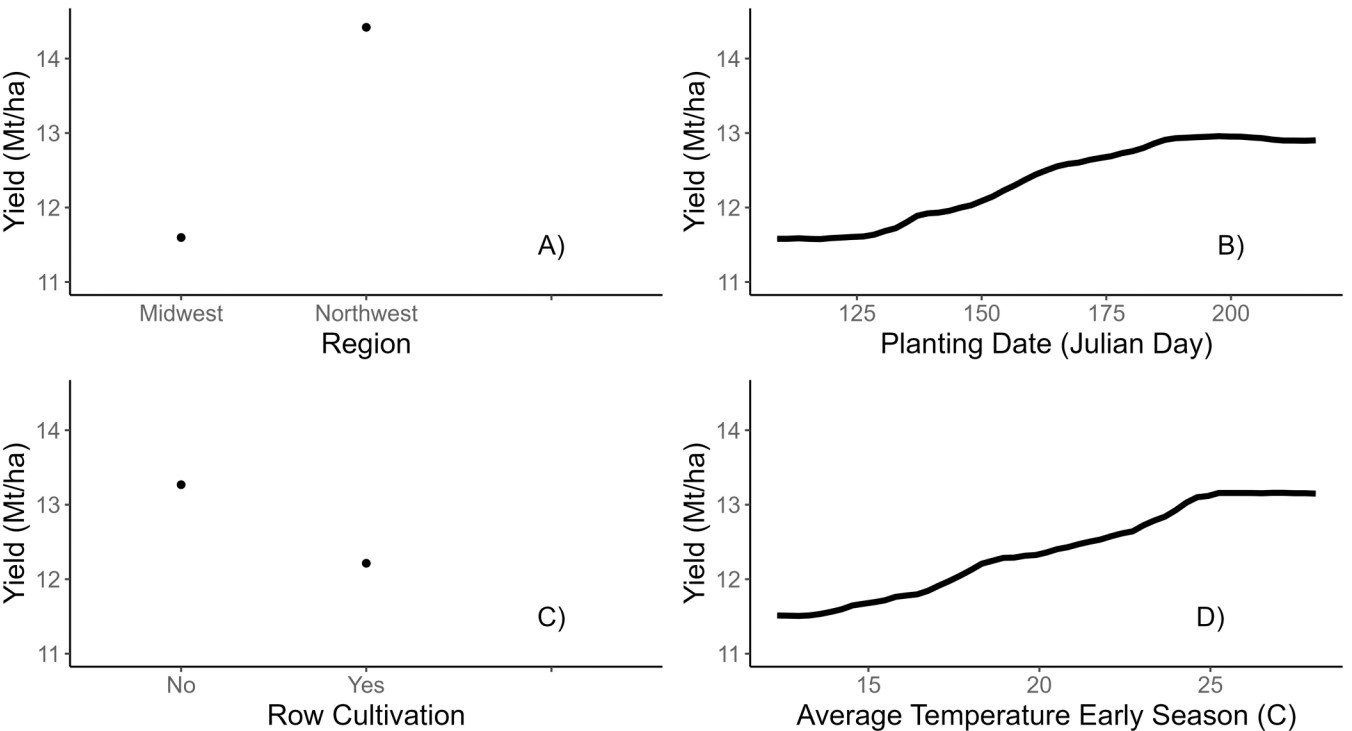

**Fig 2. Partial dependence plots of the marginal effect of 4 predictor variables displaying the most % permutation importance in the crop yield random forest model (Fig 1A).**

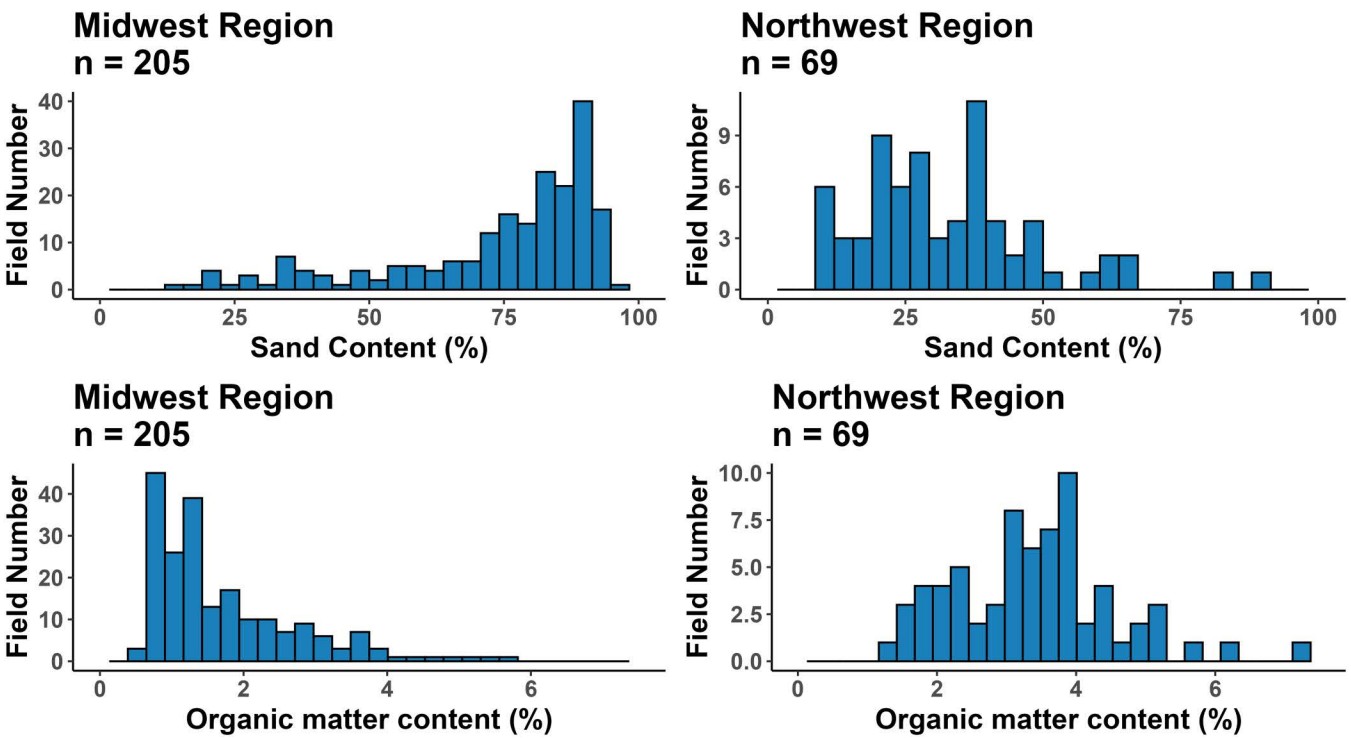

**Fig 3. Histograms representing the distribution of soil sand and organic matter content across the Midwest and Northwest regions.**

The general observed trend for PDN was that later plantings were associated with higher yields (Fig 2B). Fields planted early in the season, namely mid-April through May, were lower yielding than later planting dates. Fields planted later in the season benefitted from higher AVGT1 which resulted in higher crop yields as these two variables were positively correlated (Kendall correlation = 0.6). Even though the plantings were at the time of the year when most crops are susceptible to drought stress (July), surveyed snap bean fields had the benefit of being regularly irrigated. Similar observations were noted by others [31,32] and attributed the outcomes to an increased number of flowers and pods per plant in later plantings. Snap bean fields planted earlier in the season also were subjected to higher average temperatures (~1.5 °C) at the time of anthesis and pod setting compared to the fields planted later, which can result in flower abortion and pod abscission. Higher temperatures early in plant growth and development, as expected in midseason plantings, were likely the main reason for higher yields [33]. As snap bean is unaffected by day length (photoperiod insensitive), they can develop substantial biomass due to these higher temperatures. While the trend of higher crop yield with later planting dates is indeed observed, the conclusion that later dates would bring on greater yields cannot be made with full confidence. Processing facilities are limited by the number of tons per day they can process; therefore, growers are required to plant across a wide range of dates to spread incoming harvest loads to a manageable level each day (authors, personal observation).

AVGT1 exceeding 15°C predicted an increase in crop yield until ~22°C when the crop yield stabilizes at an average of ~12 Mt ha$^{-1}$ (Fig 2C). Higher temperatures during crop establishment allow for more vigorous snap bean growth and development, which is essential for maximizing crop yield later [31]. Warmer temperatures also facilitate seed germination and early season seedling growth, resulting in a crop that is more competitive with emerging weeds [2].

Row cultivation was associated with a ~0.5 Mt ha$^{-1}$ decrease in crop yield (Fig 2D). Row cultivation can reduce crop population density, especially early in the growing season when plants are young and sensitive and could be easily cut by

the cultivation equipment [3]. Colquhoun et al. [34] speculated that yield reductions from row cultivation vary with the type of cultivation equipment. Losses in crop population density also can happen when interrow cultivation causes excessive root damage (authors, personal observation). Row cultivation is a common practice in snap bean production as it is an effective weed control tool; however, the results suggest that cultivation can be damaging to crop yield. Future inquiries into maximizing the selectivity of row cultivation are needed.

### Residual weed density model

Based on the previous work [19], the weed community structure varied among regions. Three main weed species were identified as common and troublesome in processed snap bean production: common lambsquarters (*Chenopodium album* L.), amaranth species (*Amaranthus spp.* L.), and large crabgrass [*Digitaria sanguinalis* (L.) Scop.]. Management practices played a more prominent role in predicting mean weed density per field compared to environmental factors. The random forest model identified RC as the most important predictor of weed density, followed by STN (Fig 1B).

Row cultivation decreased weed density by ~1.0 plant $m^{-2}$ (Fig 4A). Row cultivation is a common practice in snap bean production and should be timed correctly with irrigation, as cultivating immediately before or after irrigation can result in ineffective weed control [4,35]. Interestingly, most cultivation was utilized in the MW where the highest weed densities were observed [19]. Perhaps MW fields were initially weedier than NW fields and, therefore, in greater need of row cultivation.

More spring tillage operations predicted an increase in weed density up to ~4 plants $m^{-2}$ compared to none (Fig 4B). While spring tillage is a useful management strategy, it can easily backfire with deeper plowing as many weed seeds from deeper soil layers could be brought to the surface [35]. A stable trend in weed density observed beyond three spring tillage

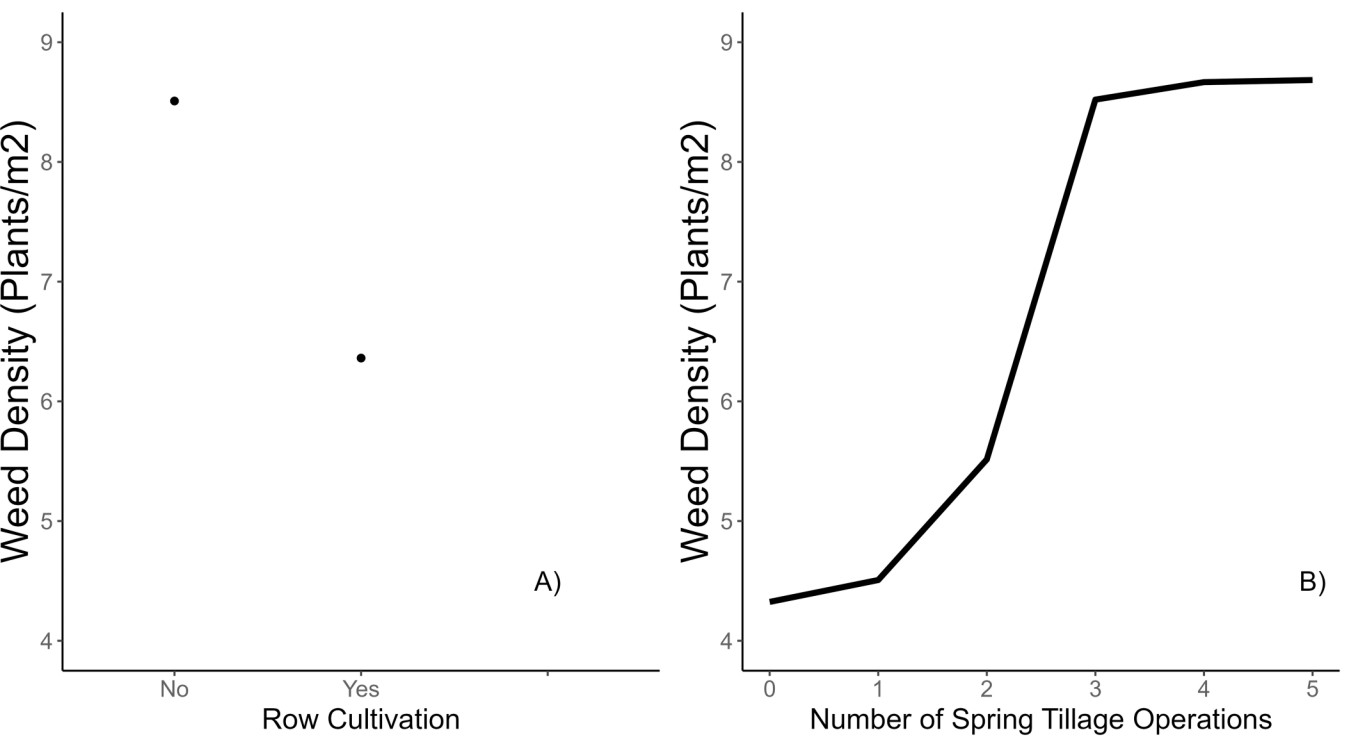

**Fig 4. Partial dependence plots of the marginal effect of the 2 predictor variables displaying the most % permutation importance in the weed density random forest model (Fig 1B).**

operations is possibly due to the exhaustion of the soil of germinable weed seeds. Soil disturbance in spring is important for the germination and establishment of many summer annual broadleaf weeds (such as common lambsquarters), concentrating emergence early in the season [36]. Use of no-till planting can decrease weed density up to ~80% when herbicides are applied. However, tillage is a necessary practice in the production of high-value crops such as snap bean and will likely continue to be used by the growers as it helps in offsetting the irregularities of non-consistent planting depth, seed-to-soil contact, or complete seed-row closure. Another possible explanation for higher weed density with more spring tillage operations is that those fields were weedier, hence the growers used multiple operations to control the weeds.

Other variables were less important in predicting weed density compared to RC and STN. Planting date displayed a trend of later plantings having fewer weeds compared to earlier plantings, possibly because the germinable weed seed fraction in the soil gets depleted by midseason plantings. However, there was an increase in weed population with the latest plantings as winter annuals start germinating. Regional differences showed that weed density was highest in the MW, which coincides with previous observations concerned with weed community [19]. Herbicides (MoA) were not a strong predictor of weed density, an implication of this might be that weed control on snap bean fields was effective overall. Growers may have relied more on the use of other practices like RC and STN for effective weed control, with the herbicide use adjusted according to the observed weed population (i.e., more MoAs on weedier fields and less on less weedier fields). Residual weeds may have been more result of mechanical weed control practices, rather than ineffective herbicide use, as PRE herbicides may have effectively controlled weeds in the early season. However, residual weeds are a cumulative result of different practices applied and what escaped control, so it cannot be stated with full certainty that only one control practice was fully responsible for weed survival late into the growing season.

## Conclusions

This survey utilized 358 snap bean production fields across three major regions where processed snap bean is grown. This is the first report of using field-level crop data and advanced data analytics to leverage insights into commercial snap bean production. The crop yield random forest model predicted that region was the most important variable in affecting snap bean yield. The region is a combination of many different factors and their interactions and the interaction of these factors (irrigation, well drained soils, low disease incident etc.) in the NW proved to have had a positive effect on snap bean yield. Planting dates were variable and showed the trend that later planting dates resulted in higher yields. They were positively correlated with higher early season temperatures, therefore higher temperature during early growth also resulted in higher yield. Planting date cannot be influenced by growers entirely and is affected more by processing facilities than anything else. The use of row cultivation resulted in a yield decrease, possibly because of the damage to the crop which is especially sensitive to this practice earlier in the growing season. However, the use of row cultivation resulted in a decrease in weed population, therefore it is a beneficial weed control practice. Further inquiry into what would be the most favorable timing or equipment type that would lead to the least crop damage, but most favorable weed control would be beneficial.

## Supporting information

**S1 File. SNAP_Data.csv.** This is the.csv file that includes the data used for analysis.
(CSV)

## Acknowledgments

We would like to thank students and employees of the Marty Williams Lab who assisted with data collection, including Mr. Nicholas Hausman, Mr. Jim Moody, Dr. Ana Saballos, and Mr. Yudai Takenaka. The authors also deeply appreciate the vegetable processors and farmers who participated in this research. Mention of a trademark, proprietary product, or

vendor does not constitute a guarantee or warranty of the product by the U.S. Department of Agriculture and does not imply its approval to the exclusion of other products or vendors that also may be suitable.

## Author contributions

**Conceptualization:** Pavle Pavlovic, Martin M. Williams II.

**Data curation:** Pavle Pavlovic.

**Formal analysis:** Pavle Pavlovic, Christopher A. Landau, Nicolas F. Martin, Martin M. Williams II.

**Funding acquisition:** Martin M. Williams II.

**Investigation:** Pavle Pavlovic, Jed B. Colquhoun, Nicholas E. Korres, Christopher A. Landau, Rui Liu, Carolyn J. Lowry, Ed Peachey, Barbara Scott, Lynn M. Sosnoskie, Mark J. VanGessel, Martin M. Williams II.

**Methodology:** Pavle Pavlovic, Martin M. Williams II.

**Project administration:** Pavle Pavlovic, Martin M. Williams II.

**Resources:** Jed B. Colquhoun, Nicholas E. Korres, Rui Liu, Carolyn J. Lowry, Ed Peachey, Barbara Scott, Lynn M. Sosnoskie, Mark J. VanGessel, Martin M. Williams II.

**Supervision:** Martin M. Williams II.

**Validation:** Christopher A. Landau, Nicolas F. Martin, Martin M. Williams II.

**Visualization:** Pavle Pavlovic.

**Writing – original draft:** Pavle Pavlovic.

**Writing – review & editing:** Jed B. Colquhoun, Nicholas E. Korres, Christopher A. Landau, Rui Liu, Carolyn J. Lowry, Nicolas F. Martin, Ed Peachey, Barbara Scott, Lynn M. Sosnoskie, Mark J. VanGessel, Martin M. Williams II.

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
