## [Decision Letter · Decision Letter 0]

22 Apr 2025

PONE-D-25-08867Associations among weed communities, management practices, and environmental factors in U.S. snap bean (Phaseolus vulgaris) production

PLOS ONE

Dear Dr. Pavlovic,

Thank you for submitting your manuscript to PLOS ONE. After careful consideration, we feel that it has merit but does not fully meet PLOS ONE’s publication criteria as it currently stands. Therefore, we invite you to submit a revised version of the manuscript that addresses the points raised during the review process.

In the introduction section should include reports on successful management of weeds in snap bean production. Also, the authors should explain why a ML random forest approach was selected?

We look forward to receiving your revised manuscript.

Kind regards,

Sumita Acharjee

Academic Editor

PLOS ONE

Journal Requirements:

4. Thank you for stating the following financial disclosure: [This research was funded by U.S. Department of Agriculture–Agricultural Research Service project 5012-12220-010-000D (“Resilience of Integrated Weed Management Systems to Climate Variability in Midwest Crop Production Systems”). Any opinions, findings, conclusions, or recommendations expressed in this publication are those of the authors and do not necessarily reflect the views of the U.S. Department of Agriculture. The mention of trade names or commercial products in this publication is solely for the purpose of providing specific information and does not imply recommendation or endorsement. The U.S. Department of Agriculture is an equal opportunity provider and employer.]. 

6. Please include captions for your Supporting Information files at the end of your manuscript, and update any in-text citations to match accordingly. Please see our Supporting Information guidelines for more information: http://journals.plos.org/plosone/s/supporting-information .

Reviewers' comments:

Reviewer's Responses to Questions

**Comments to the Author**

1. Is the manuscript technically sound, and do the data support the conclusions?

Reviewer #1: Yes

Reviewer #2: Yes

2. Has the statistical analysis been performed appropriately and rigorously? 

Reviewer #1: Yes

Reviewer #2: Yes

3. Have the authors made all data underlying the findings in their manuscript fully available?

Reviewer #1: Yes

Reviewer #2: Yes

4. Is the manuscript presented in an intelligible fashion and written in standard English?

Reviewer #1: Yes

Reviewer #2: Yes

5. Review Comments to the Author

Reviewer #1: This manuscript presents a valuable investigation into the complex interplay between weed communities, management practices, and environmental factors in US snap bean production. The use of machine learning (random forest) to analyze large-scale field survey data represents a robust methodological approach, and the findings offer practical insights for optimizing weed control and yield outcomes. However, several critical areas require improvement to strengthen the study’s validity, interpretability, and alignment with journal standards:

1. While random forest excels in capturing nonlinear relationships, compare its performance against simpler statistical models (e.g., linear regression, GLM) to contextualize its advantages. Present metrics such as R², RMSE, or AIC to justify the choice of machine learning.

2. Acknowledge that your findings are based on mechanized, high-input systems in three US regions. Discuss the limitations to scalability, particularly for smallholder or organic systems where tillage practices, herbicide use, and climate stressors differ. Highlight opportunities to adapt these insights to diverse contexts (e.g., integrating machine learning with low-tech monitoring tools in resource-limited settings).

Specific Comments:

Line 126-131: Why aggregate precipitation into 3 intervals? Justify this choice statistically/methodologically.

Line 132: "machine learning algorithm random forest". Specify "random forest regression" for clarity, as machine learning encompasses diverse methods.

Line 284-293: You attribute NW’s higher yield to "interaction of factors." Elaborate on these factors (e.g., soil moisture, pest pressure).

Line 299-300: The conclusion overstates climate impacts without projecting future scenarios. Recommend adding a brief discussion on how your findings inform adaptive strategies.

Fig 2A. The Northwest (NW) outperforms the Midwest (MW). Could soil type (e.g., sandy vs. clay) or climate adaptation (e.g., heat tolerance) explain this difference?

Reviewer #2: This manuscript presents an analysis on the relationships between management and environmental factors of snap bean fields with crop yield and weed density across the three production regions in the United States collected from survey data using random forest modelling. The authors found region to be the most important predictor of crop yield followed by planting date and whether the producer used row cultivation. Row cultivation and the number of spring tillage operations were the most important predictors of weed density. The introduction does not present a compelling story and could use some reworking. First, it would be beneficial to mention the problem this study addresses earlier rather than in the final paragraph. At first it seems this manuscript will be about shifts in management practices by snap bean producers as the first paragraph does not even mention weeds (or any pest) and most details concern weed impact on bean yields in general and effects of climate change on snap bean and weed management. I think there needs to be greater emphasis on the why here. This is not the first study to link environmental and management practices to weed density and crop yield; have others been successful? Further, why did the authors choose a ML random forest approach? Has this been used to model weed density and crop yield in the past? The methodology and analysis is appropriate for the research question yet no information is provided on why this particular approach was selected. I.e. why random forest modeling and why not another decision tree modeling approach? The results support the conclusions and will be of interest to a wide range of readers. With several small additions I believe this manuscript will be a valuable contribution to PLOS.

L92-94 – How were producers and fields selected for this survey? If it was vegetable processors who were contacted would there be a worry about potential bias?

L100 – If fields were surveyed near harvest time and surveys happened from June to Oct does this mean snap beans are continuously harvested for 5 months? Did the authors survey different regions during different times? I see confounding issues here with sampling over what is essentially the entire growing season.

L104-106 – How were quadrats placed in fields? Were they equally spaced? How did the authors ensure they were capturing variability across the field?

L105 – What is a species group? How did the authors decided which to pool and which not too?

L124-125 – What physical and chemical properties?

L143 – How was weed cover % determined?

L152 – Scaled how?

L153 – Scaled and transformed how?

L175-178 – I think this is important background material missing from the introduction.

L183-186 – What about the authors data?

L198 – On weed populations?

L234 – I think it is obvious different cultivation equipment can cause varying levels of crop damage.

L248 – Is that decline really only 1 plant/m2? It looks closer to 2 but this figure is not easy to interpret. Would a decline in weed density of 1 plant/m2 for row cultivation make it worth it? This seems very low to me.

L297 – Is a 1 plant/m2 decline in weed density a beneficial practice when this is accompanied by yield loss?

All figures are all very low resolution. Please fix.

Figure letter captions are inconsistent, some in top left others in top right, others missing entirely (Fig 3)

Fig 2A, Fig 2C and Fig 4A - are these correct? They only two dots each. Perhaps a box and whisker plot would be better to show distribution of yield and weed density by region and with and without cultivation.

Fig 3 – What is the y-axis on these?

What is the y-axis on Fig 4B? Is it weed density as well? Is this a regression line or just connecting non-continuous points? Further, there is no measure of variability presented.

6. PLOS authors have the option to publish the peer review history of their article (what does this mean? ). If published, this will include your full peer review and any attached files.

**Do you want your identity to be public for this peer review?** For information about this choice, including consent withdrawal, please see our Privacy Policy .

Reviewer #1: No

Reviewer #2: No

---

## [Author Response · Author response to Decision Letter 1]

30 Jun 2025

I didn't fully understand the guidelines regarding this point for author-generated code. I have provided more detail in my Response to Reviewers document on this point. Please let me know how to approach this point further.

---

## [Decision Letter · Decision Letter 1]

20 Aug 2025

PONE-D-25-08867R1Associations among weed communities, management practices, and environmental factors in U.S. snap bean (Phaseolus vulgaris) productionPLOS ONE

Dear Dr. Pavlovic,

Thank you for submitting your manuscript to PLOS ONE. After careful consideration, we feel that it has merit but does not fully meet PLOS ONE’s publication criteria as it currently stands. Therefore, we invite you to submit a revised version of the manuscript that addresses the points raised during the review process.With the reviewers' comments, the revised manuscript is improved compared to the previous version. However, the manuscript needs minor revision, which are indicated below.Please ensure that your decision is justified on PLOS ONE’s publication criteria  and not, for example, on novelty or perceived impact.

We look forward to receiving your revised manuscript.

Kind regards,

Sumita Acharjee

Academic Editor

PLOS ONE

Journal Requirements:

Reviewers' comments:

Reviewer's Responses to Questions

**Comments to the Author**

1. If the authors have adequately addressed your comments raised in a previous round of review and you feel that this manuscript is now acceptable for publication, you may indicate that here to bypass the “Comments to the Author” section, enter your conflict of interest statement in the “Confidential to Editor” section, and submit your "Accept" recommendation.

Reviewer #2: (No Response)

2. Is the manuscript technically sound, and do the data support the conclusions?

Reviewer #2: Yes

3. Has the statistical analysis been performed appropriately and rigorously? 

Reviewer #2: Yes

4. Have the authors made all data underlying the findings in their manuscript fully available?

Reviewer #2: Yes

5. Is the manuscript presented in an intelligible fashion and written in standard English?

Reviewer #2: Yes

6. Review Comments to the Author

Reviewer #2: Thank you for addressing nearly all of my comments. One point though regarding two comments.

L104-106 – How were quadrats placed in fields? Were they equally spaced? How did the authors

ensure they were capturing variability across the field?

We have actually provided a reference [19] in L112 to our previously published work and how the

data was collected in those surveys regarding weed populations.

L105 – What is a species group? How did the authors decided which to pool and which not too?

We have actually provided a reference [19] in L112 to our previously published work and how the

data was collected in those surveys regarding weed populations.

Despite this information being available in another published manuscript it is critical for understanding the manuscript under consideration. I think these details should not be omitted.

7. PLOS authors have the option to publish the peer review history of their article (what does this mean? ). If published, this will include your full peer review and any attached files.

**Do you want your identity to be public for this peer review?** For information about this choice, including consent withdrawal, please see our Privacy Policy .

Reviewer #2: No

---

## [Author Response · Author response to Decision Letter 2]

3 Sep 2025

Response to Requirements and Reviewers for Manuscripts PONE-D-25-08867

NOTE: The line numbers in my comments (in red) are following the line numbering of the revised manuscript without tracking changes. The line numbering of the comments from reviewers follows the line numbering pattern of the original manuscript submission.

L105-109 – How were quadrats placed in fields? Were they equally spaced? How did the authors ensure they were capturing variability across the field?

I addressed this comment previously and this is what I originally mentioned:

“We have actually provided a reference [19] in L112 to our previously published work and how the data was collected in those surveys regarding weed populations.”

But if we are just focusing on these lines, I have used the methodology described by Thomas (1985) [reference 21]. Where we stayed away from the edges of the field and aimed at spacing out our quadrats as evenly as we could, but we still accounted for weed distribution of the whole field area as best as we could. As we considered if there was consistent patchiness of some weed species, as we didn’t just go for some random single patch that may have been there in an entirety of the field area. Therefore, our statement in using the methodology from Thomas (1985) but with some slight modifications such as considering the weed distribution of the field. Therefore, we could say that there was some randomization in how we placed the quadrats and collected data, but we did somewhat account for representativeness, so it isn’t full randomization. Meaning there is some inherent bias in data sampling, but it is based on agronomic expertise and experience. We have adjusted the lines for clarification.

L109 – What is a species group? How did the authors decide which to pool and which not too?

Added a reference to our previous work [19] in how we decided for species grouping. Which also goes into more detail in how we did the actual sampling regarding the weed community data.

---

## [Editor Report · Decision Letter 2]

8 Sep 2025

Associations among weed communities, management practices, and environmental factors in U.S. snap bean (Phaseolus vulgaris) production

PONE-D-25-08867R2

Dear Dr. Pavlovic,

We’re pleased to inform you that your manuscript has been judged scientifically suitable for publication and will be formally accepted for publication once it meets all outstanding technical requirements.

Kind regards,

Sumita Acharjee

Academic Editor

PLOS ONE
---

## [Editor Report · Acceptance letter]

PONE-D-25-08867R2

PLOS ONE

Dear Dr. Pavlovic,

I'm pleased to inform you that your manuscript has been deemed suitable for publication in PLOS ONE. Congratulations! Your manuscript is now being handed over to our production team.

Kind regards,

on behalf of

Dr. Sumita Acharjee

Academic Editor

PLOS ONE